# IMPERCEPTIBLE BLACK-BOX ATTACK VIA REFINING IN SALIENT REGION

## ABSTRACT

Deep neural networks are vulnerable to adversarial examples, even in the black-box setting where the attacker only has query access to the model output. Recent studies have devised successful black-box attacks with high query efficiency. However, such performance often comes at the cost of the imperceptibility of adversarial attacks, which is essential for attackers. To address this issue, in this paper we propose to use segmentation priors for black-box attacks such that the perturbations are limited in the salient region. We find that state-of-the-art black-box attacks equipped with segmentation priors can achieve much better imperceptibility performance with little reduction in query efficiency and success rate. We further propose the Saliency Attack, a new gradient-free black-box attack that can further improve the imperceptibility by refining perturbations in the salient region. Experimental results show that the perturbations generated by our approach are much more imperceptible than the ones generated by other attacks, and are interpretable to some extent. Furthermore, our approach is found to be more robust to detection-based defense, which demonstrates its efficacy as well.

## 1 INTRODUCTION

Deep neural networks (DNNs) have achieved significant progress in wide applications, such as image classification (Deng et al., 2009), face recognition (Parkhi et al., 2015), object detection (Redmon et al., 2016), speech recognition (4, 2012) and machine translation (Bahdanau et al., 2015). Despite their success, deep learning models have revealed vulnerability to adversarial attacks (Szegedy et al., 2014). Crafted by adding some small perturbations to benign inputs, adversarial examples (AEs) can fool DNNs into making wrong predictions, which is a critical threat especially for some security-sensitive scenarios such as autonomous driving (Sun et al., 2020).

Based on the knowledge of target model, the adversarial attacks could be divided into white-box attack and black-box attack. White-box attacks (Szegedy et al., 2014; Goodfellow et al., 2015; Papernot et al., 2016; Carlini & Wagner, 2017; Moosavi-Dezfooli et al., 2016) have full access to the architecture and parameters of the target model, and can easily generate a successful adversarial example via back propagation. By contrast, black-box attacks (Narodytska & Kasiviswanathan, 2016; Ilyas et al., 2018; 2019; Moon et al., 2019; Andriushchenko et al., 2020; Li et al., 2021; Papernot et al., 2017; Liu et al., 2017) can only query the target model to obtain the output prediction, which is a more realistic and challenging setting. Since many real-world online application programming interfaces (APIs) have time or monetary limit for user query (Ilyas et al., 2018), most current research efforts focus on how to design query-efficient attacks (Ilyas et al., 2019; Moon et al., 2019; Andriushchenko et al., 2020), which indeed achieve a huge improvement. For example, so far the state-of-the-art black-box attack (Andriushchenko et al., 2020) could succeed in untargeted attack on ImageNet dataset (Deng et al., 2009) with only tens of queries on average. However, this success has sacrificed the imperceptibility of AEs, whose global perturbations are generated with various tricks like random vertical stripes (Andriushchenko et al., 2020), and thus very obvious and perceptible to human eyes (see Figure 1 (e) and (f) for examples). In effect, imperceptibility is essential for attackers. Currently online APIs usually integrate detectors into their services to detect anomalous inputs (Li et al., 2021). The images with too visible perturbations are difficult to pass these detectors, not to mention human judgement.

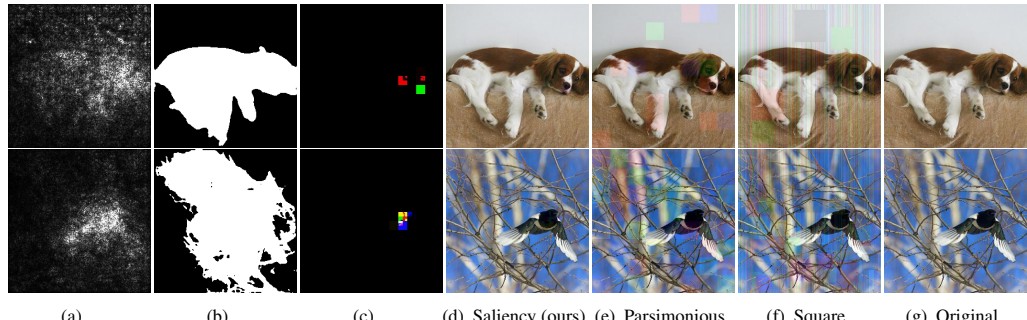

|     |     |     |     |     |     |     |
| --- | --- | --- | --- | --- | --- | --- |
| (a) | (b) | (c) | (d) Saliency (ours) | (e) Parsimonious | (f) Square | (g) Original |

Figure 1: Illustration of our Saliency Attack and current SOTA gradient-free black-box attacks. (a) BP saliency maps, specific to a given image and the corresponding class; (b) binary salient masks, generated by saliency object segmentation and roughtly accord with the region of the salient pixels in BP saliency maps; (c) perturbation, produced by refining in salient regions of (b); (d) final AEs of our Saliency Attack; (e-f) AEs of SOTA black-box attacks (Moon et al., 2019; Andriushchenko et al., 2020) whose perturbations are visible to human eyes; (g) original image.

To alleviate or even address this problem, we could utilize some priors to lead black-box attacks to search in small but efficient space, instead of global perturbations. Previous DNNs visualization work (Simonyan et al., 2014) proposed a type of saliency map to indicate which features should influence the output significantly. This BP saliency map [1] is generated by calculating the derivatives of the model output with respect to input. Just as show in Figure 1.(a), the brighter pixels in BP saliency maps denote they have a greater impact on the model output. The classical white-box attack JSMA (Papernot et al., 2016) just constructs such an adversarial saliency map and iteratively selects pixels from this map to perturb. But due to the black-box setting, we cannot derive this saliency map directly. Nevertheless, we could find that the region of bright pixels roughly represents the position of the main object in an image, which also accords with our intuition. Inspired by this, we propose to first generate a saliency mask to extract the regions of the main object in an image via image segmentation, and then limit the perturbation search in this smaller but more important regions, as presented in Figure 1.

Several recent studies also propose to generate similar local perturbations but applied in different attack settings with different methods (Dong et al., 2020; Xiang et al., 2021). In particular, Dong et al. (2020) produce saliency maps with the class activation mapping (CAM) (Zhou et al., 2016), which requires the internal information of the model and needs to change the model architecture. This is infeasible in black-box scenarios. Xiang et al. (2021) replace CAM with Grad-CAM (Selvaraju et al., 2017), which does not need to change the architecture of the model. However, both CAM and Grad-CAM still rely on the knowledge of the model, but in black-box setting the internal information of the target model is inaccessible. Thus, we choose salient object segmentation (Zhao & Wu, 2019), which can automatically extract salient object(s) in an image (see Figure 1.(b)). Compared with CAM and Grad-CAM, it doesn't need any other information except for the input image.

On the other hand, these studies (Dong et al., 2020; Xiang et al., 2021) are applied to transfer-based attacks (actually a grey-box setting), which first build a substitute model to approximate the target model, then use gradient-based white-box attacks to generate adversarial examples, and finally transfer them to attack the target model. For transfer-based attack, we must either train a substitute model from scratch with huge number of queries (Papernot et al., 2017) or assume a pretrained substitute model is trained on the similar data distribution with the target model's (Liu et al., 2017). And for the latter, the transferability of both adversarial examples and saliency maps highly depends on the selection of the pair of substitute model(s) and target model. Hence, in this paper we focus on gradient-free black-box attacks (Narodytska & Kasiviswanathan, 2016; Moon et al., 2019; Andriushchenko et al., 2020) that take no account of gradients and search for successful perturbations only according to the model output.

We add these segmentation priors to SOTA black-box attacks and find the imperceptibility of AEs is indeed enhanced but still limited due to "global perturbation" in local region (See Figure 4 and 5). To

---

[1]To distinguish the saliency map generated by the salient object segmentation model below, here we call it as BP saliency map to indicate it is generated via back propagation.

further improve the imperceptibility, our another novelty is that even in the salient region, "saliency" could still be refined. Our inspiration also comes from some DNNs visualization works (Zhou et al., 2016; Zeiler & Fergus, 2014; Olah et al., 2018). For instance, given a dog image, CAM produces a localization heatmap and shows the dog face region is most highly lighted (Zhou et al., 2016). Zeiler & Fergus (2014) systematically cover up different portions of a dog image, and finds when the dog face is obscured, the activity in the feature map and classifier output changes dramatically. In further, by combining internal feature visualization with output prediction, Olah et al. (2018) reveal even in dog's face region, its ears and eyes seem to be more important when distinguishing dogs. Therefore, we are inspired to assume the salient region in an image is progressive with respect to its impact on model output. If we could find smaller but more salient region, the perturbation will be more efficient and meanwhile the imperceptibility of AEs could also be enhanced. Thus, we propose our Saliency Attack, a new gradient-free black-box attack via refining perturbations in the salient region according to their saliency.

Our main contributions can be summarized as follows:

- To our best knowledge, we are the first to use salient object segmentation to extract binary salient masks in black-box settings. Experiments show that the SOTA black-box attacks limited in such regions can achieve much better imperceptibility performance with little reduction in query efficiency and success rate.

- We propose a new gradient-free black-box attack via refining in salient regions. Compared with the search methods used in other gradient-free attacks, our method is able to generate smaller but effective perturbations, which is interpretable to some extent and can further improve the imperceptibility.

- We demonstrate that the perturbations generated by our Saliency Attack is more robust against some detection-based defense like Feature Squeezing.

## 2 RELATED WORK

### 2.1 BLACK-BOX ATTACKS

**Gradient estimation attacks.** Gradient estimation attacks first estimate the gradients by querying the target model and then apply them to run white-box attacks (Goodfellow et al., 2015; Carlini & Wagner, 2017; Madry et al., 2018). ZOO attack (Chen et al., 2017) first adopts the symmetric difference quotient to approximate the gradients and then perform Carlini-Wagner (CW) white-box attack (Carlini & Wagner, 2017). AutoZOOM (Tu et al., 2019) uses a random vector based gradient estimation to estimate gradients, which reduces the query number per iteration from $2D$ in ZOO to $N$+1 ($D$ is the dimensionality and $N$ is the sample size). To further enhance query efficiency, Ilyas et al. (2019) propose the "tiling trick" that updates a square of pixels simultaneously instead of a single pixel, which dramatically decreases the dimensionality by a factor of $k^2$ ($k$ is tile length).

**Gradient-free attacks.** Gradient-free attacks take no account of gradients and directly generate AEs with random search or heuristic methods according to the query result. Su et al. (2019) propose one pixel attack that adopts differential evolution algorithm to perturb the most important pixel in the image. Alzantot et al. (2019) propose GenAttack, which use genetic algorithm to generate AEs. To improve the query efficiency, Moon et al. (2019) consider a discrete surrogate optimization problem that transforms the original constraint of a continuous range $[-\epsilon, +\epsilon]$ to a discrete set $\{-\epsilon, +\epsilon\}$, achieving a huge reduction in the search space. This is motivated by linear program (LP) where the optimal solution is attained at an extreme point of the feasible set (Schrijver, 1998). Combing tiling trick (Ilyas et al., 2019) and discrete optimization (Moon et al., 2019), Square Attack (Andriushchenko et al., 2020) has obtained the best result on success rate and query performance so far with a randomized search scheme.

**Hybrid attacks.** Hybrid attacks try to combine multiple types of methods, using candidate adversarial examples generated on substitute model as starting points for further optimization with gradient estimation attacks or gradient-free attacks. Suya et al. (2020) first proposed to generate candidate AEs by white-box projected gradient descent (PGD) attack (Madry et al., 2018), and further optimize them with gradient estimation methods. Wang et al. (2020) combines a white-box attack with a

gradient-free method using microbial genetic algorithm (Harvey, 2009). Similar with transfer-based attacks, hybrid attacks also suffer from the similarity between substitute model(s) and target model.

## 2.2 RELATED WORK ON IMPERCEPTIBILITY

The imperceptibility of AEs is vital for attackers and some attacks consider it in different manners. Guo et al. (2019) difine the imperceptibility as the smoothness in the sense of low frequency and thus search for AEs in frequency domain. While Zhang et al. (2020) regard the imperceptibility as the smoothness of visual content in an image and integrate Laplacian smoothing into optimization. Boundary Attack (Brendel et al., 2018) starts from an example that is already adversarial and approaches the original image gradually to reduce the distortion. Instead of global perturbations, some studies (Dong et al., 2020; Xiang et al., 2021) consider local perturbations to reduce the square of perturbations.

On the other hand, how to estimate the imperceptibility of AEs is still a problem. Most of the adversarial attacks use $L_p$ norms ($L_0$, $L_2$ and $L_\infty$) to measure the human perceptual distance between the perturbed image and the original one. Nonetheless, $L_p$ norms are found not suitable enough for human vision system (Sharif et al., 2018). To find a proper metric, Fezza et al. (2019) design subjective experiments to obtain the subjective scores on different AEs, and then test various image fidelity assessment (IFA) metrics including $L_p$ norms. Among them, most apparent distortion (MAD) (Larson & Chandler, 2010) metric is found closest to subjective scores. Hence, in this paper we adopt MAD as our main imperceptibility metric to estimate the imperceptibility of AEs.

## 3 PROPOSED METHOD

In this section, we first introduce the problem formulation of crafting adversarial examples for image classification models, and then detail our approach. The overall flowchart of our proposed Saliency Attack is depicted in Figure 2.

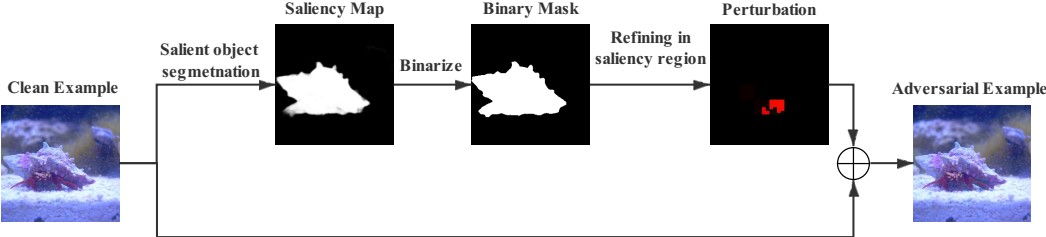

Figure 2: The overall flowchart of Saliency Attack.

## 3.1 PRELIMINARY

Given a well-trained DNN classifier $F : [0, 1]^d \rightarrow \mathbb{R}^K$, where $d$ is the dimension of the input $x$, and $K$ is the number of classes. We denote $F_k(x)$ as the predicted score that $x$ belongs to class $k$. So the classifier assigns the class that maximizes $F_k(x)$ to the input $x$. The goal of an untargeted attack is to find an adversarial example $x_{adv}$, that results in the model misclassification from the ground-truth class $y$, and meanwhile keeps the distance between the adversarial and benign input smaller than a threshold $\epsilon$:

$$\arg\max_{k=1,\ldots,K} F_k(x_{adv}) \neq y, \quad s.t. \ ||x_{adv} - x||_p \leq \epsilon, \ x_{adv} \in [0, 1]^d \quad (1)$$

where the other constraint indicates that the generated $x_{adv}$ must be an image in valid range. In this paper we focus on $L_\infty$ as our distance norm following (Moon et al., 2019; Andriushchenko et al., 2020).

Traditionally, this task of finding $x_{adv}$ can be rephrased as solving a constrained continuous problem:

$$\max_{x_{adv} \in [0,1]^d} L(F(x_{adv}), y), \quad s.t. \ ||x_{adv} - x||_\infty \leq \epsilon \quad (2)$$

where $L$ is a loss function. To improve the query efficiency, Moon et al. (2019) transform the continuous problem into a discrete surrogate problem, where the perturbation $\delta = x_{adv} - x$ is

generated at the corner of the $L_\infty$ ball. Besides, we propose to limit $\delta$ only in the salient region. So our complete optimization problem is:

$$\max_{x_{adv}\in[0,1]^d} L(F(x_{adv}),y), \quad s.t. \ ||\delta||_\infty \in \{-\epsilon,+\epsilon\}, \ \delta \in \mathcal{S} \tag{3}$$

where $\mathcal{S}$ is the set of the pixels in salient region.

## 3.2 SALIENT OBJECT SEGMENTATION

Salient object segmentation, as one changing task in image segmentation domain, aims to automatically and accurately extract salient object in an image. Specifically, given an input image, salient object segmentation model can generate a saliency map where the brighter pixels denote they have higher saliency scores, just as the saliency map shows in Figure 2. Compared with other model visualization methods like CAM (Zhou et al., 2016) and Grad-CAM (Selvaraju et al., 2017), this type of model does not require any information other than the input image, which is very suitable for the black-box setting. We adopt the Pyramid Feature Attention (PFA) network (Zhao & Wu, 2019) that achieves the state-of-the-art performance in multiple datasets. After obtaining the saliency map where all pixel values are between 0 and 1, we use a binarization threshold $\phi$ to transform the saliency map to a binary salient mask. The binarization can be expressed as

$$s_{i,j}^* = \begin{cases} 0 & s_{i,j} < \phi \\ 1 & s_{i,j} \geq \phi \end{cases} \tag{4}$$

where $s$ is the saliency map and $s^*$ is the binary salient mask, $s_{ij}$ and $s_{i,j}^*$ are the corresponding value at the position $i,j$. With this saliency mask $s^*$, we could limit the perturbation search in the salient region.

## 3.3 REFINING PERTURBATION IN SALIENT REGION

In this part, we design a search algorithm to refine perturbations based on a tree structure for an image by using the idea of depth-first search. As shown in Figure 3, an input image merged with its saliency mask is split into four initial blocks (a coarse grid). Then we try to add a $+\epsilon$ or $-\epsilon$ perturbation on each initial block individually and find the best block that maximizes the loss function for further perturbation. After finding the best one among all initial blocks, we can further split this block into smaller blocks (a finer grid), and again try to add a perturbation on each block individually (at this time we just flip the perturbation for convenience e.g. $+\epsilon$ to $-\epsilon$) to find the best smaller block. We iterate this process until the minimal block (e.g. 1 pixel) or no smaller block has a better loss. Then we go back to the last level of split blocks and use the second best block for further perturbation.

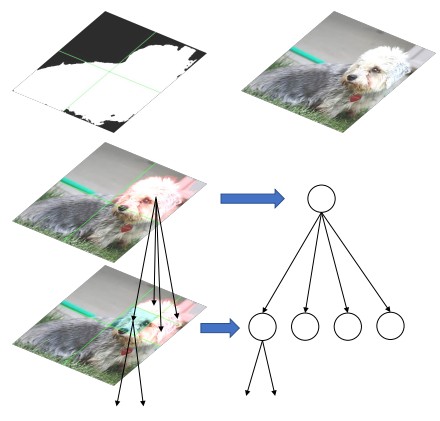

We use the loss function from CW attack (Carlini & Wagner, 2017) for untargted attack:

$$L(x) = -\max(Z(x_{adv})_o - \max_{i \neq o}(Z(x_{adv})_i), \ 0) \tag{5}$$

Figure 3: The illustration of refining on a tree structure corresponding to the image blocks. Among initial blocks, each block is a root node of a tree, and its child nodes are the finer blocks in the current block.

where $Z(x_{adv})_o$ is the logit with respect to the true class of the original image. In this way, the loss function is imposed to leave a margin between the true class and other classes. The refining process is presented in Algorithm 1, and the overall saliency attack is given in Algorithm 2.

## 4 EXPERIMENTS

In this paper, we evaluate the performance of the Saliency Attack comparing against Boundary Attack (Brendel et al., 2018), TVDBA (Li & Chen, 2021) Parsimonious attack (Moon et al., 2019),

---

**Algorithm 1:** Refine

---
**Input:** block $b$, block size $k$
**Output:** adversarial example $x_{adv}$

1  $k'$ is the block size of $b$;
2  $\{b_1, b_2, ..., b_n\} \leftarrow$ split $b$ into $(k'/k)^2$ finer blocks;
3  $\{b_{\pi(1)}, b_{\pi(2)}, ..., b_{\pi(n)}\} \leftarrow$ sort $\{b_1, b_2, ..., b_n\}$ in descending order according to $L(b)$, which is calculated by using Eq. 5;
4  **for** *each block* $e \in \{b_{\pi(1)}, b_{\pi(2)}, ..., b_{\pi(n)}\}$ **do**
5     **if** $L(e) > \hat{L}$ **then**
6         $x_{adv} \leftarrow x + \delta \cup \{e\}$;
7         $\hat{L} \leftarrow L(e)$;
8         **if** $k > 1$ **then**
9             $k \leftarrow k/2$;
10            Recursively call $x_{adv} \leftarrow$ Refine$(e, k)$;
11         **end**
12     **end**
13 **end**

---

**Algorithm 2:** Saliency Attack

---
**Input:** original image $x$, initial block size $k_{int}$, query budget
**Output:** adversarial example $x_{adv}$

1  $\delta \leftarrow \emptyset$ is the perturbation;
2  **while** $k_{int} > 1$ *and not exceeding query budget* **do**
3     Running Algorithm 2: $x_{adv} \leftarrow$ Refine$(x, k_{int})$;
4     $k_{int} \leftarrow k_{int}/2$;
5 **end**

---

Square attack (Andriushchenko et al., 2020) and their modified version equipped with segmentation priors. Among them, Boundary Attack is able to constantly reduce the distortion to a very small magnitude, TVDBA tries to minimize the distortion via integrating Structural SIMilarity (SSIM) (Wang et al., 2004) into the loss, and Square Attack can achieve the current state-of-the-art query efficiency and success rate in the black-box setting. We consider $L_\infty$ threat model on ImageNet dataset (Deng et al., 2009), set $\epsilon$ to 0.05 in [0,1] scale, and test 1000 randomly selected exmaples in all experiments. The threshold $\phi$ to produce binary salient masks is chosen to be 0.1. All parameters of the compared attacks remain consistent with those recommended in their papers. We use Inception v3 (Szegedy et al., 2016) as the target model, and different query budgets $\{3000, 10000, 30000\}$ for untargeted attack. For performance metrics, we employ commonly used success rate (SR) and average number of queries (Avg. queries). To evaluate the imperceptibility of AEs, we consider $L_0$, $L_2$ and MAD (see the appendix for details), which is validated as closest to human vision system among existing IFA metrics (Fezza et al., 2019). All these three imperceptibility metrics are the smaller the better.

### 4.1 SOTA BLACK-BOX ATTACKS WITH SEGMENTATION PRIORS

To verify the feasibility of our segmentation priors, we modify Parsimonious Attack and Square Attack so that the perturbations are limited in salient regions. We show the results in Table 1 and some examples in Figure 4. From the examples, we can easily find that the perturbations of original attacks are very obvious in the entire image due to global perturbations, while their segmentation versions are relatively more imperceptible since no perturbation exists in background regions.

Besides, through quantitative analysis, the attacks equipped with segmentation priors can indeed achieve much better imperceptibility performance with around one third improvement in $L_2$ and MAD metric. $L_0$ is also restricted to less than $50\%$, which implies the average square of salient regions among all examples only occupies less than half of the whole image. And this improvement only comes at little cost in SR and Avg. queries, which is still in the same order of magnitude. We also present the change of MAD scores in Figure 9 in appendix. Thus, the results demonstrate that the segmentation prior is indeed able to enhance the imperceptibility of other black-box attacks.

Table 1: Comparison of black-box attacks with their modified version with segmentation priors.

| Method | Avg. queries | SR | $L_2$ | $L_0$ | MAD |
|---|---|---|---|---|---|
| Parsimonious | 739 | 98.2% | 53.20 | 100.0% | 22.17 |
| Parsimonious-seg | 828 | 96.7% | 45.64 | 42.70% | 13.95 |
| Square | 222 | 99.7% | 59.76 | 99.03% | 25.36 |
| Square-seg | 563 | 97.1% | 40.45 | 41.89% | 16.10 |

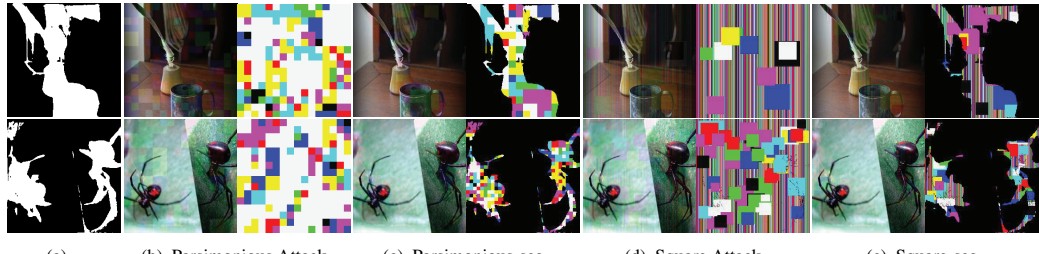

(a)  (b) Parsimonious Attack  (c) Parsimonious-seg  (d) Square Attack  (e) Square-seg

Figure 4: Examples of SOTA black-box attacks compared with their modefied version with segmentation priors. (a) binary salient mask; (b-e) pairs of AEs and their perturbation.

## 4.2 COMPARISON OF SALIENCY ATTACK WITH OTHER ATTACKS

Although the segmentation prior is helpful to the imperceptibility, the perturbations generated by other attacks are still global, taking up almost all salient regions. Hence, we design our Saliency Attack via further refining perturbations in salient regions. Original image are resized to $256 \times 256$ and we set the initial block size to 16. For an imperceptible AE, a low MAD value is important. So besides SR, we use a new metric $SR_{true}$. It denotes the rate of successful AEs whose MAD values are below some threshold. Here we choose 30 as the threshold because we find the AEs with $MAD \leq 30$ are basically imperceptible to human eyes (See Figure 10 in appendix for details).

The comparison results are given in Table 2. We can find in all query budgets, our Saliency Attack achieves outstanding performance with a huge gap in terms of $SR_{true}$ and three imperceptibility metrics. Although TVDBA, Parsimonious and Square Attack can obtain a better SR, their successful AEs are not true imperceptible, and will be easily detected by some defenses or humans. For Boundary Attack, it can gradually make progress on $SR_{true}$, $L_2$ and MAD as the query budget increases, but they are still much worse than Saliency Attack. This is because it takes at least hundreds of thousands of queries for Boundary Attack to converge (Brendel et al., 2018), which is infeasible in practical. We further test more eamples to verify the statistical significance in Table 5, and present a comparison of $SR_{true}$ versus query number in Figure 11 in appendix.

From the exhabited examples in Figure 5, we can find the AEs of Boundary Attack contain obvious coarse textures because of inadequate query budget. In addition, even restricted in the same salient regions, the perturbations of Parsimonious-seg and Square seg attack are complex and irregular, while our perturbations are smaller and more important, roughtly corresponding to the bright pixels in BP saliency map. They further represent the postions of dogs' noses or ears, which accord with our inspiration before.

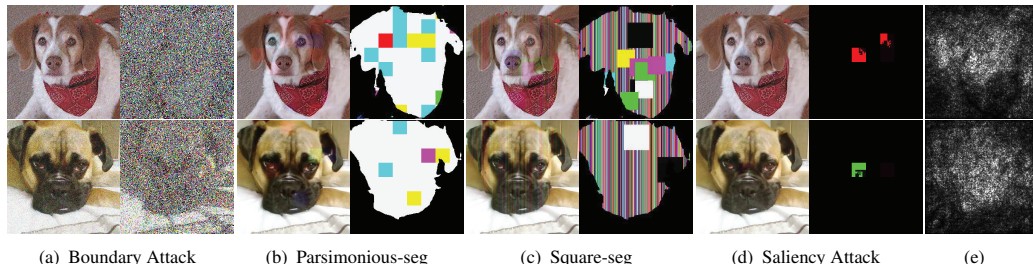

(a) Boundary Attack  (b) Parsimonious-seg  (c) Square-seg  (d) Saliency Attack  (e)

Figure 5: Examples of our Saliency Attack and other compared attacks. (a-d) pairs of AEs and their perturbation; (e) BP saliency maps.

Table 2: Comparison of different attacks. $SR_{true}$ denotes the rate of successful AEs with $MAD \leq 30$, which are believed imperceptible to human eyes.

| Method | Query Budget | SR | $SR_{true}$ | $L_2$ | $L_0$ | MAD |
|---|---|---|---|---|---|---|
| Boundary | 3000 | 100.0% | 13.6% | 68.13 | 99.6% | 84.96 |
| | 10000 | 100.0% | 30.2% | 58.82 | 99.7% | 71.61 |
| | 30000 | 100.0% | 46.0% | 58.30 | 99.8% | 62.77 |
| TVDBA | 3000 | 90.1% | 22.9% | 10.77 | 19.7% | 39.20 |
| | 10000 | 96.9% | 23.2% | 10.96 | 20.5% | 39.42 |
| | 30000 | 98.4% | 23.0% | 11.06 | 20.7% | 39.66 |
| Parsimonious | 3000 | 91.9% | 14.0% | 22.17 | 100.0% | 52.01 |
| | 10000 | 98.2% | 14.1% | 22.17 | 100.0% | 53.20 |
| | 30000 | 99.8% | 14.1% | 22.17 | 100.0% | 53.36 |
| Parsimonious-seg | 3000 | 89.8% | 10.8% | 14.01 | 42.7% | 45.50 |
| | 10000 | 96.7% | 12.0% | 13.95 | 42.7% | 45.64 |
| | 30000 | 99.7% | 12.4% | 13.92 | 42.7% | 45.48 |
| Square | 3000 | 98.4% | 2.2% | 25.36 | 99.0% | 59.29 |
| | 10000 | 99.7% | 1.9% | 25.36 | 99.0% | 59.76 |
| | 30000 | 99.9% | 2.7% | 25.35 | 99.0% | 58.24 |
| Square-seg | 3000 | 89.5% | 19.9% | 16.26 | 42.7% | 40.41 |
| | 10000 | 97.1% | 22.5% | 16.10 | 41.9% | 40.45 |
| | 30000 | 98.7% | 25.5% | 16.02 | 41.7% | 39.05 |
| Saliency (ours) | 3000 | 85.6% | **80.2%** | **3.52** | **3.3%** | **12.32** |
| | 10000 | 93.6% | **86.2%** | **3.71** | **3.8%** | **12.88** |
| | 30000 | 96.1% | **87.9%** | **3.89** | **4.3%** | **13.28** |

## 4.3 HYPERPARAMETER TUNING

Our Saliency Attack contains only one hyperparameter namely the initial block size $k$ which determines the first level of split blocks. We test the effect of different $k$ on SR, imperceptibility and query number respectively in Figure 6. As $k$ decreases, SR and imperceptibility can be improved and SR reaches the peak when $k$ equals 16. This is because with smaller initial blocks, Saliency Attack can search for perturbations more finely and accurately leading to higher SR and better imperceptibility performance. Meanwhile inevitably more queries are needed, especially for sorting initial blocks. That is why under a limited query budget like 10000, a turning point occurs in SR.

We also show some examples in Figure 7. It can be observed that as k decreases, the generated perturbations also become smaller but more salient. For instance, in the first row the perturbation with $k = 128$ roughly covers the region of the dog face, which is also the brightest region in the BP saliency map. While the perturbation with $k = 32$ or $k = 16$ focuses on smaller region of the dog ear. This indicates that our Saliency Attack could find smaller and more important perturbations progressively as the initial block size decreases. And the perturbations are interpretable to some extent like perturbing the specific regions of dog's ears, eyes or nose.

## 4.4 ABLATION STUDY

We do ablation study of Saliency Attack, including refining in salient region, in non-salient region and without saliency (refining in the whole image). We also design a greedy search as baseline to demonstrate our Refine search. We test multiple block sizes for greedy search and use 32 as the best choice (see Table 6 in appendix). The results and some examples are given in Table 3 and Figure 8. Notice that refining in salient region and refining without saliency generate the same or almost the same perturbations, which means the salient regions indeed contain useful parts and enhance the query efficiency by limiting the search space. But for refining in non-salient region, its perturbation is more complex and visible with worse query efficiency and SR. Compared with greedy search, our Refine search has much better query efficiency and $SR_{true}$, which proves its superiority. Hence, we can conclude that both salient region and Refine search facilitate our Saliency Attack.

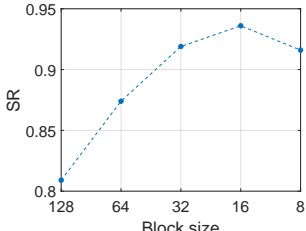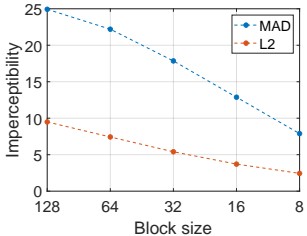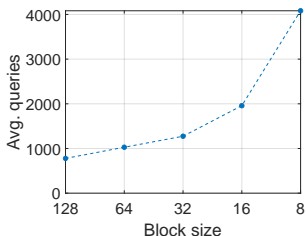

Figure 6: The effect of initial block size in SR, imperceptibility ($L_2$ and MAD) and avg. queries.

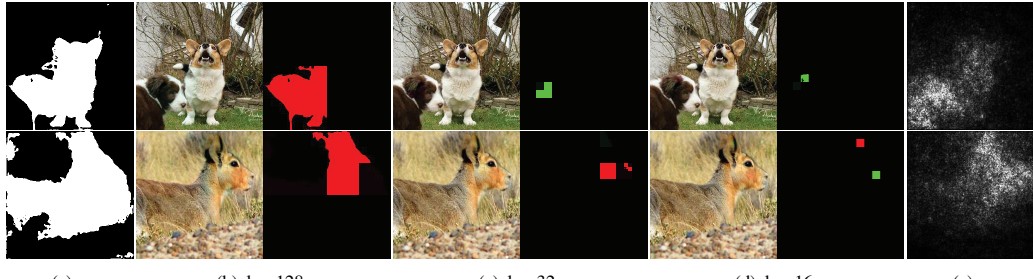

| (a) | (b) k = 128 | (c) k = 32 | (d) k = 16 | (e) |

Figure 7: Examples of DFS-seg attack with different initial block sizes. (a) binary salient mask; (b-d) exmaples of Saliency Attack with different initial block sizes; (e) BP saliency map.

Table 3: Ablation study of Saliency attack under 10000 query budget.

| Method | Avg. queries | SR | $SR_{true}$ | $L_2$ | $L_0$ | MAD |
|---|---|---|---|---|---|---|
| refine in salient region | **1958** | 93.6% | **86.2%** | **3.71** | **3.8%** | **12.88** |
| refine in non-salient region | 3128 | 78.2% | 57.0% | 4.94 | 6.5% | 21.58 |
| refine without saliency | 2563 | 95.5% | 79.6% | 3.84 | 4.2% | 16.35 |
| greedy search in salient region | 2727 | 56.0% | 50.7% | 4.37 | 4.7% | **12.87** |

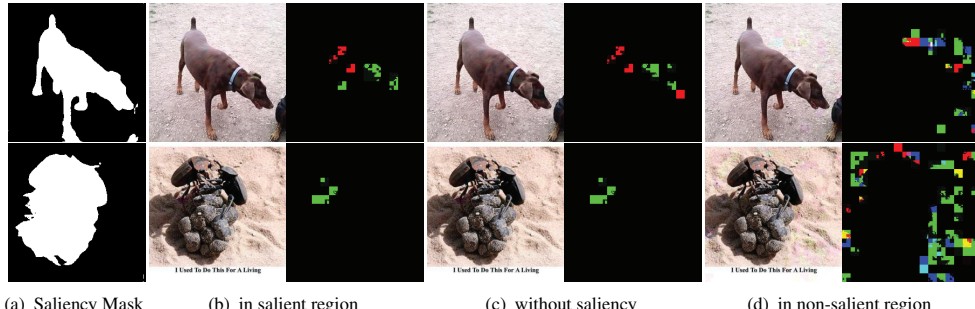

| (a) Saliency Mask | (b) in salient region | (c) without saliency | (d) in non-salient region |

Figure 8: Examples of Saliency Attack with different strategies.

## 4.5 ATTACKING DETECTION-BASED DEFENSE

To further validate the superiority of our method, we attack against one detection-based defense Feature Squeezing (Xu et al., 2018), which applies transformations to input image and judge whether it is adversarial through the stability of model output. We consider the detection rate of both successful adversarial examples (SAEs) and failure adversarial examples (FAEs). This is because if we could detect FAEs, we can raise the alarm that some potential attacker is attacking the model. From Table 4, we can find our Saliency attack is able to achieve lower detection rate in both SAEs and FAEs than other attacks, which proves the imperceptibility of our AEs from the defensive perspective.

Table 4: Results for attacking against detection-based defense

| | Parsimonious | Square | Boundary | Saliency |
|---|---|---|---|---|
| Detection rate (SAEs) | 33.3% | 47.8% | 61.5% | **21.2%** |
| Detection rate (FAEs) | 14.5% | 16.7% | \ | **11.7%** |

## 5 CONCLUSION

In this paper, we propose to provide segmentation priors for black-box attacks to improve the imperceptibility of adversarial examples, whose perturbations are limited in salient regions. We utilize a salient object segmentation model to produce such saliency maps with no need for any information other than the input image. Experiments show that SOTA black-box attacks equipped with segmentation priors can indeed achieve better imperceptibility performance. Furthermore, we devise a new gradient-free black-box Saliency attack that further enhances the imperceptibility via refining in salient regions. The perturbations generated are much smaller, imperceptible and interpretable to some extent. Finally, we also attack against some detection-based defense and the results show that the adversarial examples generated by our attack are harder to be detected.

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

## A APPENDIX

### A.1 MAD METRIC

Most Apparent distortion (MAD) metric (Larson & Chandler, 2010) is one of the state-of-the-art full-reference image quality assessment methods. MAD attempts to merge two separate strategies for two kinds of distorted images respectively. For high-quality images with near-threshold distortion (just visible), MAD focuses on detection-based strategy to look for distortions in the presence of the image. While for low-quality images with suprathreshold distortion (clearly visible), MAD focuses on appearance-based strategy to look for image content in the presence of the distortions. MAD will control the weight of two strategies according to the type of distorted images. The calculation process of MAD can be summarized as following steps and we recommend interested readers to read the original literature.

1. Compute locations of visible distortions based on luminance images.
2. Combine the visibility map with local error image.
3. Decompose both the distorted and original images into log-Gabor subbands.
4. Calculate different statistics of each subband.
5. Calculate the adaptive blending score.

### A.2 SCATTER PLOT OF MAD SCORES

Equipped with segmentation priors, 68.1% and 94.3% examples of Parsimonious-seg attack and Square-seg attack have better MAD scores compared with their original version respectively. Since Parsimonious Attack uses a greedy local search while Square attack adopts a random search, it is obvious that limiting the search in salient regions is more helpful to Square Attack.

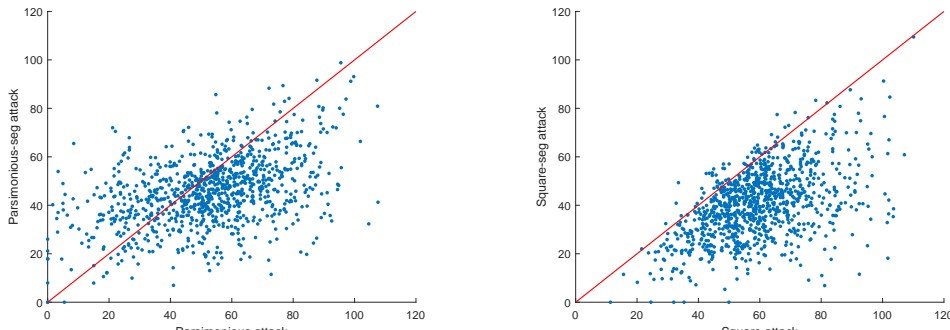

Figure 9: MAD scores of Parsimonious Attack vs. Parsimonious-seg attack and Square Attack vs. Square-seg Attack.

### A.3 THRESHOLD FOR MAD METRIC

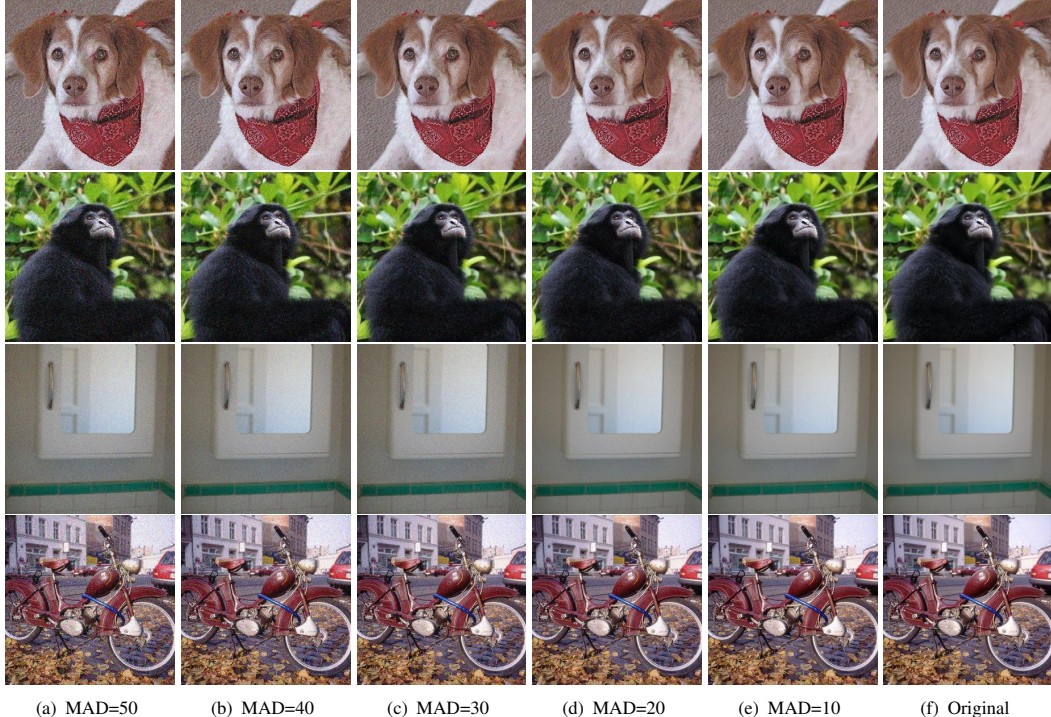

| (a) MAD=50 | (b) MAD=40 | (c) MAD=30 | (d) MAD=20 | (e) MAD=10 | (f) Original |
|---|---|---|---|---|---|

Figure 10: Adversarial examples generated by boundary attack with different MAD scores. We can find the threshold $MAD \leq 30$ is roughly enough to indicate an imperceptible adversarial example.

### A.4 TRUE SUCCESS RATE VERSUS QUERY NUMBER PLOT

Here we further present a comparison of $SR_{true}$ versus query number under different MAD thresholds. It shows that Square Attack and Parsimonious Attack can generate some successful AEs with

very few queries due to global perturbations and some tricks like initializing perturbations with random vertical stripes (Andriushchenko et al., 2020), but they lack the ability to further improve the imperceptibility. Instead, Saliency Attack conservatively select small regions to perturb, hence the query efficiency is a little lower than Square Attack and Parsimonious Attack at the beginning but soon exceeds them.

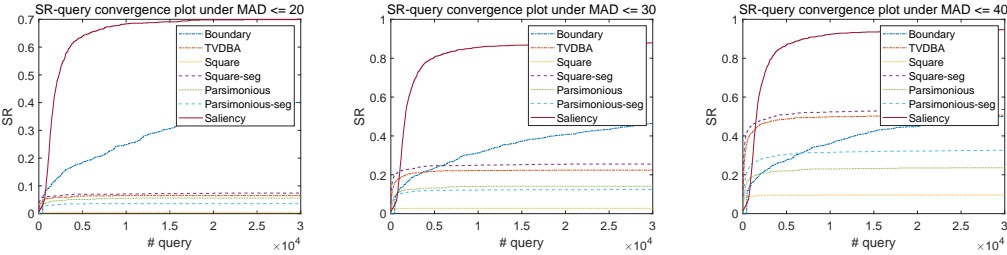

Figure 11: Queries vs. True Success Rate under different thresholds of MAD scores

## A.5 COMPARISON OF SALIENCY ATTACK WITH OTHER ATTACKS

We randomly choose 10,000 examples from the ImageNet validation set and divide them into 10 groups. Then we test different attacks under 3000 query budget and report their mean and standard deviation (SD). The best results are recorded in bold based on Wilcoxon signed-rank test with significance level at 0.05. We can find that our Saliency Attack is significantly better than all the baselines.

Table 5: Comparison of Saliency Attack with other attacks under 3000 query budget.

| Method | SR $\pm$ SD | SR$_{\text{true}}$ $\pm$ SD | L$_2$ $\pm$ SD | L$_0$ $\pm$ SD | MAD $\pm$ SD |
|---|---|---|---|---|---|
| TVDBA | 90.1%±0.011 | 23.4%±0.015 | 10.64±0.153 | 19.0%±0.005 | 37.65±3.631 |
| Parsimonious | 92.4%±0.007 | 14.7%±0.008 | 22.17±0.000 | 100.0%±0.000 | 51.80±0.115 |
| Parsimonious-seg | 88.9%±0.010 | 11.4%±0.014 | 13.54 ±0.497 | 35.5%±0.026 | 45.31±0.212 |
| Square | 98.3%±0.003 | 2.7%±0.004 | 25.34 ±0.021 | 99.0%±0.001 | 57.29±0.045 |
| Square-seg | 87.5%±0.014 | 20.3%±0.022 | 14.71 ±0.526 | 34.5%±0.028 | 38.56±0.538 |
| Saliency (ours) | 84.3%±0.010 | **79.8%±0.009** | **3.44±0.054** | **2.9%±0.001** | **12.07±0.198** |

## A.6 GREEDY SEARCH IN SALIENT REGION WITH DIFFERENT BLOCK SIZES

We test different block sizes for greedy search in salient region, and choose the best one to be compared in ablation study.

Table 6: Greedy search in salient region with different block sizes.

| Block size | Avg. queries | SR | SR$_{\text{true}}$ | L$_2$ | L$_0$ | MAD |
|---|---|---|---|---|---|---|
| 128 | 57 | 20.6% | 18.4% | 7.32 | 12.8% | 11.50 |
| 64 | 512 | 37.4% | 31.3% | 6.37 | 10.2% | 15.18 |
| 32 | 2727 | 56.0% | **50.7%** | 4.37 | 4.7% | 12.87 |
| 16 | 4039 | 35.4% | 35.2% | 1.79 | 0.8% | 4.84 |

