# OpenReview forum: "Imperceptible Black-box Attack via Refining in Salient Region"
_ICLR.cc/2022/Conference — ICLR 2022 Submitted_

### Official Review · Reviewer_Hs6q · 2021-11-01

**Correctness:** 3
**Technical Novelty And Significance:** 3
**Empirical Novelty And Significance:** 2
**Recommendation:** 5
**Confidence:** 4

**Main Review:**

The topic of imperceptible black box is very interesting and not well studied, the ideas of generating perturbing blocks in salient regions are interesting, however, this paper can be improved with more sufficient justifications on motivations and experiments. Moreover, its technical contribution is not very high.

The motivation of generating the perturbed regions in the salient object region is not sufficiently justified: salient object can still be mostly a
smooth low frequency area, e.g., smooth human face/skin, and hence generating the perturbation in this area may not be reasonable. Moreover, it is not clear whether generating perturbation in salient object area makes more sense than generating perturbations in the non-smoothness, high frequency, or  heavy texture area.

The experiment part is not totally convincing. 1) There is only one data set, which is 1000 images from ImageNet. However, what these images look like will affect the results a lot. For example, does most salient objects have heavy texture area? 2) Also, there is no comparison to the related method in approach of Zhang (2020). 3) I can not find the information of which classification network is used in the experiments. Are multiple networks used in the experiments?

This paper used an existing salient object segmentation approach, and so its major technical contribution is the tree based method to generate perturbation blocks, which is however a little straightforward and ad-hoc.

**Summary Of The Paper:**

This paper proposed a method to generate imperceptible attack in black box attack scenario by generating local perturbation blocks in salient regions. It used salient object segmentation to obtain the salient region, then applied a tree search method to find smallest blocks within the salient region that can cause the maximal change in predicted class logits. Experiments on 1000 Imagenet examples are conducted, compared to several existing baselines, showing that the proposed method can improve achieve more imperceptible attacks (where imperceptibility is measured by metric MAD).

**Summary Of The Review:**

Overall, I liked the idea of this paper, and enjoyed reading it. However, it needs better justification on motivations, as well as improvement on experiments.

---

> ### Author Response · Authors · 2021-11-23
> **Response to Reviewer Hs6q**
>
> We thank the reviewer for insightful comments.
> >**Q1 About the motivation to use salient region**
>
> **A1** Indeed, the salient region is not perfect and the perturbations added to some smooth low frequency area in salient region may be obvious and visible. That’s exactly the reason why we devise our search algorithm to further refine the perturbations in salient region, to make the perturbations as small as possible. From the results and some examples (Figure 5, Table 2), we can find the final perturbations of our attack only occupy a small part in the salient region, and this is in line with the original purpose of the algorithm design. We also test more experiments with 10,000 examples and find our Saliency attack can outperform other attacks stably and the result is statistically significant. We will consider smoothness, high frequency, or heavy texture area in future work.
>
> >**Q2 About the tested images**
>
> **A2** We agree with you that the type of images may affect the result, but not significantly. As suggested by reviewer zKra (in A3), we have tested 10,000 examples and the small standard deviation reflects our attack can work stably on different types of images (Table 5, page16, revised paper). Although we cannot obtain the statistics of the images with heavy texture area, it is more likely that not all of the 10,000 images from ImageNet contain heavy texture area.
>
> >**Q3  About (Zhang et al., 2020)**
>
> **A3** (Zhang et al., 2020) considers the smoothness of visual content, but their method is applied in white-box attack setting, which cannot be applied to black-box setting. Besides, as suggested by Reviewer 2cEz, we have tested (Li & Chen, 2021) which also aims to generate imperceptible perturbations. We find our method can outperform this work in all aspects (Table 2, page 8, revised paper).
>
> Li, N., & Chen, Z. (2021). Toward Visual Distortion in Black-Box Attacks. IEEE Transactions on Image Processing, 30, 6156-6167.
>
> >**Q4  About the model**
>
> **A4** In our experiments we test attacks on Inception v3 model (Line 8, para 1, page 6).
>
> >**Q5 About the novelty and contribution**
>
> **A5**  **For our search method**, as introduced in para 1 (page 3), we design it from the motivation that even in the salient region of an image, “saliency” could still be refined. For example, given a dog image, previous visualization work shows that the region of the dog’s body will influence the model’s output more significantly than other regions. Then the head in the dog’s body is obviously more important. In further, dog’s nose or ears might be even more significant. Inspired from this phenomenon, we devise our search method to divide an image into a tree structure and refine the perturbations iteratively. We aim to perturb smaller but more important regions to generate imperceptible adversarial examples. We did comprehensive experiments to justify the effectiveness of our search method. The results show that it outperforms existing search algorithms (i.e., local search and random search, in Table 2, page 8) and common baseline (i.e., greedy search, in Table 3, page 9). Besides, the examples of the generated perturbations also exhibit the “progressive saliency” of perturbations (Figure 7, page 9) which is in line with the phenomenon above. Hence, we believe our search method is effective and realized our inspiration.
>
> **For using saliency maps**, indeed, this idea has already appeared in the field of adversarial attack. However, to the best of our knowledge, we are the first to apply it to **black-box attack on image classification model**. We have reviewed two similar works (Dong et al., 2020; Xiang et al., 2021) in the paper, which use CAM/grad-CAM to generate saliency maps and can only be applied in white-box or grey-box settings. We introduced them and analyzed their difference and infeasibility to black-box setting thoroughly (para. 3,4, page 2). Besides, we want to clarify that our main contribution is not just introducing saliency maps into black-box attacks, but how to process and utilize saliency maps in black-box setting. The combination of saliency maps and our search algorithm bring out smaller and more imperceptible perturbations, which greatly alleviates the problem of imperceptibility in black-box attacks.

---

### Official Review · Reviewer_zKra · 2021-11-02

**Correctness:** 3
**Technical Novelty And Significance:** 2
**Empirical Novelty And Significance:** 3
**Recommendation:** 5
**Confidence:** 4

**Main Review:**

The paper proposes a few optimisations that can improve the imperceptibility of generated adversarial perturbations. They also demonstrate that some of the exiting black-box adversarial attacks can benefit from their optimisations. In addition to this, they also propose a search algorithm that can further narrow down the candidate regions for adversarial perturbations. They also demonstrate that adversarial examples generated by their technique have a higher success rate for evading detection mechanisms.

In the recent past, a richer class of gradient-free black-box techniques have been proposed in the literature. Why is the paper considering only two of those? For instance, why not consider techniques based on Bayesian optimisation. The same applies to methods for detecting adversarial examples. Why was Feature Squeezing chosen as a baseline to evaluate the success rate?

In addition to the above, the author(s) report the results only over a sample of 1000 images from ImageNet, why only such a small sample was considered, and I would also encourage the authors to evaluate over multiple samples & report the variance too.

**Summary Of The Paper:**

There has been a lot of interest in improving the query efficiency of black-box attacks in the recent past. However, these techniques produce examples that a human in the loop can quickly identify. The authors propose using segmentation priors to improve the black-box attacks so that the perturbations are restricted to the salient regions of the image. In addition to this, they also present a technique that improves the imperceptibility without forgoing the query efficiency.

**Summary Of The Review:**

In general, I found the paper easy to follow. I also find the direction explored by the paper to be quite promising but found the experiments & the baselines chosen by the authors are lacking. I would encourage the authors to include a diverse set of baselines. For instance, I would also demonstrate how the optimization proposed in the paper improves gradient-free black-box adversarial example generation techniques other than the chosen few. Also, evaluate the evasion rate of adversarial examples generated by the proposed technique against a diverse set of detection mechanisms.

---

> ### Author Response · Authors · 2021-11-23
> **Response to Reviewer zKra**
>
> We thank the reviewer for valuable comments and suggestions.
>
> > **Q1 Why only consider two gradient-free black-box attacks?**
>
> **A1** In this paper we actually consider three gradient-free black-box attacks, including Parsimonious Attack (Moon et al., 2019), Square Attack (Andriushchenko et al., 2020) and Boundary Attack (Brendel et al., 2018). Parsimonious Attack first proposed discrete optimization which dramatically reduces the search space and save lots of queries, which is essential in black-box attack setting. We also follow this work to use discrete optimization. Square Attack achieves the SOTA success rate (SR) and query efficiency performance in our score-based black-box attack setting. Boundary Attack can constantly reduce the distortion to a very small magnitude. Since our paper focus on imperceptible black-box attack, we also consider this one. For black-box attacks with Bayesian optimization such as (Shukla et al., 2019; Ru et al., 2019), we find they underperform Square Attack in terms of SR and query efficiency, and their distortion are also moderate. Besides, as suggested by Reviewer 2cEz, we tested (Li & Chen, 2021) which also aims to generate imperceptible perturbations. We find our method can outperform this work in all aspects (Table 2, page 8, revised paper).
>
> Shukla, S. N., Sahu, A. K., Willmott, D., & Kolter, J. Z. (2019). Black-box adversarial attacks with bayesian optimization. arXiv preprint arXiv:1909.13857.
>
> Ru, B., Cobb, A., Blaas, A., & Gal, Y. (2019, September). Bayesopt adversarial attack. In International Conference on Learning Representations.
>
> Li, N., & Chen, Z. (2021). Toward Visual Distortion in Black-Box Attacks. IEEE Transactions on Image Processing, 30, 6156-6167.
>
> >**Q2 Why choose Feature Squeezing?**
>
> **A2** The defense techniques can be roughly divided into robustness-based defenses and detection-based defenses. Robustness-based defenses, such as adversarial training, JPEG compression and gradient obfuscation, attempt to make DNNs more robust so that they can classify adversarial examples correctly. While detection-based defenses aim to detect adversarial examples in advance then the model can refuse these malicious inputs. Since our paper focus on the imperceptibility of adversarial examples, naturally we choose to attack against some detection-based defense. However, though lots of detection-based defenses exist, few of them can be applied to ImageNet dataset. In further, we find fewer works are open-source. That’s why we finally choose Feature Squeezing which is open-source and can be used for ImageNet. This defense is also well-known for detecting adversarial examples based on the uncertainty of the model outputs.
>
> >**Q3 About small number of examples tested**
>
> **A3** There are two reasons for using 1000 examples. The first is in consideration of the running time, especially for some time-consuming attack such as Boundary Attack (Brendel et al., 2018). The second is that most papers of black-box attacks test 1000 examples in their experiments, so we follow the routine.
>
> We strongly agree that using more examples will make the experiments more convincing. As suggested, we have tested more examples in the revised paper. Specifically, we randomly choose 10000 examples from the ImageNet validation set and divide them into 10 groups. For each group we ran the experiments to test the attacks. Finally, we reported the mean and standard deviation. We also conducted a Wilcoxon signed-rank test with significance level 0.05 to test whether the difference is significant. The results (Appendix 5, page 16, revised paper) show that our attack is significantly better than all the baselines.

---

### Official Review · Reviewer_3FMW · 2021-11-02

**Correctness:** 4
**Technical Novelty And Significance:** 2
**Empirical Novelty And Significance:** 3
**Recommendation:** 5
**Confidence:** 4

**Main Review:**

Strengths
- The proposed method is simple and achieves the goal of improving the imperceptibility of the perturbations: using the segmentation prior is effective for avoiding changes on the background, and the Saliency Attack attains better MAD score.

- Several ablation studies are presented to illustrate the proposed method.

Weaknesses
- The paper focuses on improving the imperceptibility of $\ell_\infty$-attacks with a fixed budget $\epsilon=0.05$, in practice reducing the number of pixels perturbed (a sort of minimization of the $\ell_0$-norm). However, I think that how visible the perturbations are is more a property of the threat model (set of feasible changes) rather than of the attack used: for example, using a smaller $\epsilon$ for the same attacks would increase their imperceptibility. Moreover, other $\ell_p$-norms (e.g. Square Attack has versions for $\ell_2$ and $\ell_1$ [A]), including $\ell_0$, or perceptual metrics like LPIPS [B] should be considered, since they produce more localized changes than $\ell_\infty$-attacks.

- There are a few works which aim at finding which areas of an image the attacker should perturb to have invisible changes, both with black- and white-box attacks [C, D]. In particular, [C] show that small perturbations in the $\ell_\infty$-norm are not necessary if limited to certain areas to the image to preserve imperceptibility.

- The presentation of the method in Algorithms 1 and 2 seems a bit confuse: first, $\delta$ is not defined. Second, the "Refine" function is called recursively on smaller blocks, but also repeatedly with smaller $k$ in Algorithm 2: is this correct? Also, in Line 9 $k$ is halved: shouldn't it be restarted from the original value when going over the next iteration of the loop in L4?

- From the images in Figure 4 it seems that for Parsimonious and Square Attack perturbations are sampled also on areas which are outside the mask given by the segmentation prior: are those candidates evaluated, spending queries of the total budget (although they can't change the current perturbations)?

[A] https://arxiv.org/abs/2103.01208
[B] https://arxiv.org/abs/2006.12655
[C] https://arxiv.org/abs/1909.05040
[D] https://arxiv.org/abs/2010.13773

**Summary Of The Paper:**

The paper studies how to reduce the perceptibility of the perturbations to the original images produced by black-box adversarial attacks (for the $\ell_\infty$-threat model). In particular, it proposes to use a prior based on segmentation techniques to localize the changes on the subject of the image and leave the background unaltered. Moreover, the Saliency Attack is introduced to further reduce the fraction of the original image which is modified to induce misclassification.

**Summary Of The Review:**

Overall, I think the proposed method is effective, but, especially given that the technical novelty is limited, it should be better positioned: if the main goal is imperceptibility of the perturbations it should be compared to other kinds of attacks beyond $\ell_\infty$ ones. Otherwise, the authors should better motivate why improving the $\ell_\infty$-attacks with such fixed threshold is relevant.

---

> ### Author Response · Authors · 2021-11-23
> **Response to Reviewer 3FMW**
>
> We thank the reviewer for insightful comments and advices.
>
> > **Q1: About other $L_p$-norms for imperceptibility**
>
> **A1**: Yes, we strongly agree that the imperceptibility is more a property of the threat model, and other $L_p$ norms and more perceptual metrics should be considered. Actually until now there is no standard definition of imperceptibility in the literature. Among $L_p$ norms, we cannot decide which one is the best. One-pixel attack (Su et al., 2019) can achieve extreme $L_0$ distance but someone may find the single perturbed pixel is salient in the image.$L_2$ attacks such as AutoZOOM (Tu et al., 2019) distribute the perturbation to the whole image but the distortion in the clean background region may be visible. Beyond $L_p$ norms, some work regard the imperceptibility as smoothness of visual content (Zhang et al., 2020) while some other work think the distortion in low frequency is imperceptible (Guo et al., 2019). (we review some work in section 2.2, page 4) Hence, the imperceptibility is still ambiguous and our work is an attempt to perturb smaller but more important regions to achieve the imperceptibility under $L_{\infty}$ norm. We will extend our method to other $L_p$ norms and test more perceptual metrics in future work.
>
> >**Q2: About the effectiveness of $L_{\infty}$ norm for imperceptibility**
>
> **A2**: As we discussed in A1, different $L_p$ norms have different features and it is hard to decide which one is the best. For [c] (Croce & Hein, 2019) which considers sparse imperceptible perturbation and uses $L_0$ as the evaluation metric, it still could not neglect the applicability of other $L_p$-norm attacks. For example, in our work, by refining the perturbations in the salient region, we can still generate imperceptible attacks using $L_{\infty}$ norm (see examples in Figure 1, 5, 7, 8, page 2, 7, 9) and the perturbations are also sparse and have very small $L_0$ value (less than 5%) in Table 2 (page 8). Thus, we believe other $L_p$ norms such as $L_{\infty}$ can also improve the imperceptibility (e.g. sparsity of perturbations).
>
> >**Q3: About the algorithm**
>
> **A3**: $\delta$ is the perturbation, which is initialized as an empty set. We have added this in the revised paper (in Line 1, Algorithm 2, page 6).
>
> As the definitions show, $k$ in Algorithm 2 is the initial block size and is halved in outer iterations, while the $k$ in Algorithm 1 is the block size of the split blocks and is halved in inner iterations. For clarification, we have modified the $k$ in Algorithm 2 to $k_{int}$. For example, the size of original image x is 256*256 and the initial block size $k_{int}$  is 128. We call Refine($x,k_{int}=128$). Then in Algorithm 1, we will use $k$ to split $x$ into finer blocks. According to recursively call Refine() or exit from inner Refine(), k will be halved or timed by 2. But the initial block size $k_{int}$ won’t change until Refine() ends and we go back to the outer iteration in Algorithm 2. Then the initial block size $k_{int}$ is halved to 64 and call Refine() again. The reason that we design such an outer iteration is we find different initial block sizes are suitable for attacking different images. Some images are easy to perturb, so a large initial block size can save lots of queries. While some other images may need elaborate perturbations to cause misclassification thus a small initial block size is more suitable. Therefore, we design this outer iteration to utilize the advantages of both large and small initial block sizes.
>
> >**Q4: About the sampling of Parsimonious-seg and Square-seg**
>
> **A4**: When we apply the binary salient masks to Parsimonious and Square Attack, we have limited their search space. In other words, they can only search or sample perturbations in salient regions. So the comparison between them and our Saliency Attack is fair. Since the perturbations of Parsimonious and Square Attack are both square-based while the salient region is not a square, the square-shaped perturbations near the edge of the salient region could be fragmentary. As a result, it may seem like that the perturbations are also sampled outside the salient region, which in fact is not true.

---

> > ### Comment · Reviewer_3FMW · 2021-11-25
> > **Post rebuttal comments**
> >
> > I thank the authors for the detailed response and the clarifications about the algorithm and the sampling scheme for the other methods.
> >
> > About imperceptibility, I agree that other norms can be used to achieve it, and that there's no established way to measure it in the context of adversarial attacks. However, in my opinion, the shortcoming in the current evaluation is that it compares the proposed Saliency Attack in terms of metric which are not optimized by (most of) the competitors: as an example, both Square Attack and Parsimonious (and segmented versions) use the whole budget wrt $\ell_\infty$ and $\ell_0$ since the first query to maximize efficiency, then they will perform poorly when compared wrt $\ell_2$ and $\ell_0$ to the proposed which starts with empty perturbations. If the authors consider e.g. $\ell_0$ as a good proxy for imperceptibility, then methods which optimize it (or produce $\ell_0$-bounded perturbations) should be include as baselines in my opinion (same for the other metrics). Even for $\ell_\infty$-attacks different values of the budget $\epsilon$ should be compared, since the visibility of the perturbations crucially depends on it.
> >
> > As further note, Croce & Hein (2019), to produce imperceptible changes, integrate $\ell_0$-attacks (not focused on query efficiency) with a mask which estimates the maximal perturbation for each element, which seems to be quite aligned with the metrics used for the experimental evaluations of the Saliency Attack. Moreover, Croce & Hein (2019) propose a black-box attack, unlike what stated in the reply to Reviewer 2cEz.
> >
> > As minor comment, it seems that in the replies to the other reviewers, the authors reference Croce & Hein (2021) for the TVDBA method, while I think it should be Li & Chen (2021).

---

> > > ### Author Response · Authors · 2021-11-27
> > > **Response to Reviewer 3FMW**
> > >
> > > Thanks very much for your detailed explanation and valuable advice.
> > >
> > > About imperceptibility, we appreciate your advice that other methods which optimize $L_2$ or $L_0$ should be considered as baselines. Among the baselines in our paper, actually Boundary Attack (Brendel et al., 2018) just optimize $L_2$ norm which starts from a successful adversarial example and then iteratively decrease the $L_2$ distance by random walking; TVBDA (Li & Chen, 2021) integrate SSIM into the loss for optimization. SSIM is an image quality assessment and some work think it is more suitable for imperceptibility than Lp norms (Sharif et al., 2018; Fezza et al. 2019). Besides, although we didn't consider a method which optimize L0 norm, our saliency attack actually achieves very outstanding $L_0$ performance. The $L_0$ of our method is only around 4% in all query budgets while other attacks are at least an order of magnitude larger than ours in Table 2 (page 8).
> > >
> > > In addition, we thank for your recommendation of Croce & Hein (2019), and we tried to add this attack into our baselines. However, we find its **One-pixel modification** step consumes too many queries in black-box setting. For 299\*299\*3 image (we consider Inceptionv3 model on ImageNet), one-pixel modification needs $299 \times 299 \times 2^{3} = 715,208$ queries, which far exceeds our maximal query budget of 30,000. This method is hard to be compared with other baselines, which usually converge with thousands of queries. Although our work mainly focus on the imperceptibility, the query efficiency is also important for black-box attacks. So we finally didn't compare this work. But we will try to consider other works which optimize $L_0$ norms in future work.
> > >
> > > We are very sorry for our typos and we have corrected them.

---

> > > > ### Comment · Reviewer_3FMW · 2021-11-30
> > > > **Further comments**
> > > >
> > > > Thanks for the further clarifications.
> > > >
> > > > About the comparison in terms of $l_0$-norm, on ImageNet it is possible to achieve around 100% of success rate with much smaller budget and limited queries (e.g. 0.1% of modified pixels, see https://arxiv.org/abs/2006.12834). This is why I consider important to add a comparison to other methods.
> > > >
> > > > Overall, I think the proposed method has some interesting aspects, and, in my opinion, a more comprehensive evaluation is needed to position it more clearly among the existing attacks.

---

> > > > > ### Author Response · Authors · 2021-11-30
> > > > > **Response to Reviewer 3FMW**
> > > > >
> > > > > Thanks very much for your comment and advices. We will consider your suggested work (Croce et al., 2020) and other $L_0$ attacks in future work.

---

### Official Review · Reviewer_2cEz · 2021-11-02

**Correctness:** 2
**Technical Novelty And Significance:** 2
**Empirical Novelty And Significance:** 2
**Recommendation:** 5
**Confidence:** 4

**Main Review:**

This work tackles an important problem, that is, developing an imperceptible black-box attack. However, there are two main issues concerning the technical approach and the experimental analysis.

- From a technical point of view the search algorithm devised to refine perturbation is exceedingly naive. It is based on an iterative procedure that resembles quadtree analysis in image processing. The first split of the salient region generates four initial blocks, and a perturbation is added to them to find those that maximize the loss function. After finding the best block, it is further split and the procedure is repeated until the block is reduced to one pixel or no smaller blocks can achieve a better loss. Then, this same procedure is applied again to the second best block, etc. As it is described, this procedure has neither a theoretical nor an experimental justification. In addition, a similar idea but applied in a different context can be found in Guo et al. (2020) and saliency maps to add adversarial attacks are also explored in Sun et al. (2021). Overall, I feel that the contribution is not significant in terms of technical approach.

- The approach is not compared with other black-box attacks that also have the objective to make the perturbation imperceptible, see references below. In fact, experiments are carried out by modifying some SOTA approaches by including saliency regions and comparing them with the proposed solution (see Table 2). In my opinion this is too limited to validate the proposal.

Typos: imperceptiblity, tunning

References
- Guo et al. Watch out! Motion is Blurring the Vision of Your Deep Neural Networks, NeurIPS 2020
- Sun et al. Generating facial expression adversarial examples based on saliency map, Image and Vision Computing 2021
- Wang et al. Perception Improvement for Free: Exploring Imperceptible Black-box Adversarial Attacks on Image Classification, arXiv 2020
- Liu et al. GreedyFool: Multi-Factor Imperceptibility and Its Application to Designing Black-box Adversarial Example Attacks, arXiv 2020
- Croce and Hein, Sparse and Imperceivable Adversarial Attacks, ICCV 2019
- Gragnaniello et al. Perceptual quality-preserving black-box attack against deep learning image classifiers, Pattern Recognition Letters 2021
- Li and Chen, Toward Visual Distortion in Black-Box Attacks, IEEE Transactions on Image Processing 2021


=========== Post rebuttal comments ===========

I appreciate that the authors better clarified their contribution and included a new comparison in the experimental section, hence I increase my score. However, I still believe that the paper needs much more comparisons with state-of-the-art and that the proposal should be better justified from a theoretical point of view.

**Summary Of The Paper:**

This paper proposes a black-box attack where, by relying on segmentation priors, the perturbation is applied only in the salient region. This allows one to obtain reduce perceptibility with a limited number of queries and a small reduction in success rate. More specifically, once the salient region has been identified, a refining procedure is carried out to find small areas where the perturbation should be added. Experiments are performed on ImageNet and results are compared with those of some SOTA methods and their variants that work on saliency regions.

**Summary Of The Review:**

In my opinion the technical novelty introduced in this paper is too limited and also its experimental validation should be improved by considering more relevant methods for comparison.

---

> ### Author Response · Authors · 2021-11-23
> **Response to Reviewer 2cEz**
>
> We thank the reviewer for the constructive and valuable comments. In below we address the raised concerns in turn.
>
> > **Q1: About the technical significance of our proposed approach**
>
> **A1**: First we notice there is a misunderstanding of our approach in this comment. After reaching the minimal block or no smaller block has a better loss, we **backtrack to the last level of split blocks** (instead of going back to the level of initial blocks and “applying the procedure again to the second best block”) and choose the best one of the remaining blocks for further perturbation. The intuition is that we always refine perturbations in a region that is as small as possible, which accords with our goal of perturbing smaller but more important regions to generate imperceptible AEs.
>
> **For our search algorithm**, we agree that it is simple. However, we think that a simple yet effective algorithm is always desirable, because it can be easily understood and implemented. For examples, Parsimonious attack (Moon et al., 2019) uses greedy local search; Square attack (Andriushchenko et al., 2020) adopts an even simpler random search. But these simple algorithms can achieve very strong performance, because they are particularly suited to the problems. For the present work, the proposed search algorithm is also suited to the problem of generating AEs with small perturbations, and the results have demonstrated its effectiveness. We appreciate for the suggestion that we need to **justify the search algorithm**. In Table 2 (page 8), Saliency significantly outperforms Parsimonious-seg and Square-seg, which indicates that our search algorithm is better than the local search algorithm in Parsimonious Attack and the random search algorithm in Square Attack. Moreover, we have added experiments to compare our search algorithm with a new baseline greedy search in salient region, which greedily perturbs blocks to maximize marginal gain. We have fine-tuned the block size of the greedy search algorithm and the comparison results are presented in Table 3 (page 9). It can be seen that our algorithm is still significantly better. The above results that our search algorithm outperforms existing search algorithms (i.e., local search and random search) and common baseline (i.e., greedy search) indicate that it is indeed effective.
>
> **For using saliency maps**, indeed, this idea has already appeared in the field of adversarial attack. However, to the best of our knowledge, we are the first to apply it to **black-box attack on image classification model**. The two papers mentioned in the comment (Guo et al.,2020; Sun et al., 2021) are both for white-box attacks. Actually we have reviewed two similar works (Dong et al., 2020; Xiang et al., 2021) in the paper, which use CAM/grad-CAM to generate saliency maps and can only be applied in white-box or grey-box settings. We introduced them and analyzed their difference and infeasibility to black-box setting thoroughly (para. 3,4, page 2). Besides, we want to clarify that our main contribution is not just introducing saliency maps into black-box attacks, but how to process and utilize saliency maps in black-box setting. The combination of saliency maps and our search algorithm bring out smaller and more imperceptible perturbations, which greatly alleviates the problem of imperceptibility in black-box attacks.
>
> >**Q2: About the comparison with other imperceptible attacks**
>
> **A2**: We thank the reviewer for providing these related papers. After going through the 7 papers, we found that among them 5 papers (Guo et al., 2020; Sun et al., 2021; Wang et al., 2020; Liu et al., 2020; Croce & Hein, 2019) are for white-box attacks or grey-box attacks, which cannot be applied to black-box setting considered in our paper. Among the left two papers, (Gragnaniello et al.,2021) has been only tested on CIFAR-10 and MCS2018 dataset (much smaller scale than ImageNet), and needs a dedicated CNN. So we did not compare with this work in the added experiments. Finally, we have added experiments comparing (Li & Chen, 2021) with our approach. The results are presented in Table 2 (page 8). It can be found that our approach outperforms (Li & Chen, 2021) in all aspects.

---

> > ### Comment · Reviewer_2cEz · 2021-11-27
> > **Thank you for your comments**
> >
> > I want to thank the authors for answering to all my questions and for carrying out further experiments.
> > I really appreciate the comparison with another method and encourage the authors to further enrich the experimental part in future work.
> > I also think that they need to strengthen the theoretical justification of their approach to make it more convincing.

---

> > > ### Author Response · Authors · 2021-11-28
> > > **Response to Reviewer 2cEz**
> > >
> > > We thank very much for your comment and advice. For theoretical justification, one main difficulty is that the search space of our algorithm is always changing due to different sizes of splitting blocks. So it is very hard to prove the performance bound. On the other hand, since we utilize the idea of depth-first search for a tree structure, actually the time complexity is just $O(N)$ ($N$ is the number of blocks), which is very straightforward. Anyway, we will try to give some theoretical justification in future work.

---

### Author Response · Authors · 2021-11-23
**Comment for all reviewers**

We would like to thank the reviewers for their comments and suggestions on the manuscript. In below we address each comment from each reviewer in turn. As well, we would like to let you know that any change in the revised paper has been marked in yellow color so that you can easily identify the changes applied to the paper.

---

### Decision · Program_Chairs · 2022-01-20

**Decision:**

Reject

**Comment:**

In this paper, the authors propose to use segmentation priors for black-box attacks such that the perturbations are limited in the salient region. They also find that state-of-the-art black-box attacks equipped with segmentation priors can achieve much better imperceptibility performance with little reduction in query efficiency and success rate. Hence, the auithors propose the Saliency Attack, a new gradient-free black-box attack, that can further improve the imperceptibility by refining perturbations in the salient region.
The reviewers think that the proposed method is simple and important, and the authors have responded properly to some comments.
However, the reviewers still are not satisfied with the experimental evaluation and comparisons, as the authors can only try to compare with other ideas and test more models in the future.
In summary, I think the manuscript at its current staus cannot be accepted.